# Driving Factors in the Development of Eye Movement Patterns in Chinese Reading: The Roles of Linguistic Ability and Oculomotor Maturation

**DOI:** 10.3390/bs15040426

**Published:** 2025-03-26

**Authors:** Meihua Guo, Nina Liu, Jingen Wu, Chengchieh Li, Guoli Yan

**Affiliations:** 1Faculty of Psychology, Tianjin Normal University, Tianjin 300387, China; gmh1509@mnnu.edu.cn (M.G.); psyygl@163.com (G.Y.); 2School of Education and Psychology, Minnan Normal University, Zhangzhou 363000, China; 3Academy of Psychology and Behavior, Tianjin Normal University, Tianjin 300387, China; 4College of Art, Yango University, Fuzhou 350015, China

**Keywords:** linguistic abilities, oculomotor maturation, eye movement patterns, reading development

## Abstract

The mechanisms driving the development of eye movement patterns is an unresolved debate in children during reading, with three competing hypotheses: the oculomotor-tuning hypothesis, the linguistic-proficiency hypothesis, and the combined hypothesis that incorporates both. This study examined eye movement patterns in 215 Chinese children from first to fifth grade using sentence-reading tasks. Oculomotor maturation was measured through saccade tasks, and linguistic abilities were assessed using Chinese character recognition and vocabulary knowledge tests. Path analysis explored how these factors predict temporal and spatial eye movement measures. Results indicated that temporal measures were primarily driven by linguistic abilities, supporting the linguistic-proficiency hypothesis. Spatial measures, however, were influenced by both linguistic abilities and oculomotor maturation, supporting the combined hypothesis. These findings diverge from predictions of the E-Z Reader model in alphabetic scripts, likely due to the unique visual complexity of Chinese characters.

## 1. Introduction

Children’s reading not only shape their cognitive and linguistic development, but also profoundly affects their academic achievements and future career prospects ([23]; [62]). Therefore, understanding how reading abilities develop in children is a critical research topic. Eye movement patterns during reading provide precise insights into real-time text processing, offering a valuable lens through which the development of children’s reading abilities can be studied ([41]; [42]; [56], [57]). Previous research on the development of children’s eye movement patterns has demonstrated that, with increasing age, fixation durations, fixation counts, and refixation rates decrease, while saccade amplitudes, skipping rates, and regressions increase. These patterns typically approach adult levels around the age of 11 ([5]; [20]; [46]; [55]). However, a fundamental question remains debated, that is, are these developmental changes primarily driven by linguistic abilities, oculomotor maturation, or a combination of both ([22]; [25]; [35]; [58]).

[58] ([58]) proposed three hypotheses to explain the driving forces behind the development of children’s eye movement patterns: the linguistic-proficiency hypothesis, the oculomotor-tuning hypothesis, and the combined hypothesis. According to the linguistic-proficiency hypothesis, advancements in linguistic abilities are the primary determinants of children’s eye movement development. As reading proficiency improves, children process lexical information more efficiently, resulting in shorter fixation durations, longer saccade amplitudes, and fewer regressions. In contrast, the oculomotor-tuning hypothesis attributes these changes to the maturation of the oculomotor system. Young children typically exhibit slower and less accurate saccades due to their underdeveloped oculomotor systems ([44]; [53]). Over time, extensive visual experience helps to finetune oculomotor control, leading to more efficient and precise eye movement patterns during reading. The combined hypothesis suggests that the development of children’s eye movement patterns results from the interplay of linguistic and cognitive skill improvements alongside the maturation of oculomotor control.

To test these hypotheses, [58] ([58]) conducted computational simulations using the E-Z Reader model, an eye movement model for adult reading based on alphabetic scripts. They used the model’s standard adult parameter settings as a baseline and systematically adjusted parameters related to linguistic abilities, such as lexical processing speed (*α*1). The simulations successfully replicated key empirical features of children’s reading eye movement patterns, showing that slower lexical processing speeds (i.e., increasing the *α*1 value in the model) were the primary drivers of longer fixation durations and higher regression rates during reading.

To investigate the role of oculomotor maturation, the researchers modified parameters associated with saccade programming (e.g., M1 and M2) within the model. However, these adjustments alone failed to reproduce children’s shorter saccade amplitudes and higher regression rates. Neither increasing systematic errors in the saccade system nor adding random noise to the parameter values significantly affected the model’s predictions of eye movement measures. These findings lend strong support to the linguistic-proficiency hypothesis, suggesting that differences in eye movement patterns between children and adults are predominantly driven by the development of linguistic processing abilities, particularly lexical processing speed. [45] ([45]) further confirmed the findings and highlighted the critical role of orthographic knowledge in shaping the development of eye movement patterns during reading.

Beyond computational simulations, earlier empirical research also supported the linguistic-proficiency hypothesis. [22] ([22]) conducted a longitudinal study with 21 German child readers, testing them at 2 time points: second and fourth grades. The experiment involved reading German sentences containing target words of varying lengths and frequencies, with total fixation duration on the target words serving as the dependent variable. Participants also completed saccade tasks (STs), which measured oculomotor maturation, and word/picture naming tasks, which assessed linguistic abilities. The results showed that oculomotor maturation in second grade did not significantly predict total fixation duration on target words in fourth grade, whereas linguistic abilities had a significant predictive effect.

Some studies on oculomotor training for children with reading difficulties provided some evidence supporting the oculomotor-tuning hypothesis ([25]). For instance, [50] ([50]) implemented a six-week fixation training program for children aged 5 to 14 who exhibited both reading difficulties and oculomotor deficits. The program involved exercises, like tracking animated targets on a computer screen ([63]). Following the training, participants showed significant gains in fixation skills and reading fluency. Other studies focus on oculomotor control training, which included oculomotor exercises, search tasks and saccade exercises also help children reduced fixation durations and faster reading speeds ([6]; [9]), While these findings suggest that effective oculomotor training can improve eye movement patterns and reading efficiency, they primarily reflect outcomes in children with reading challenges, thus providing limited evidence for the broader applicability of the oculomotor-tuning hypothesis.

Taken together, studies on alphabetic scripts provide robust evidence that linguistic ability primarily drives the development of children’s reading eye movement patterns, with comparatively less support for the role of oculomotor maturation. Unlike alphabetic scripts, Chinese is a logographic language characterized by high visual complexity, dense information content, and the absence of word boundaries ([33]; [34]; [43]). These features may necessitate more efficient visual processing during Chinese reading ([71], [70]), which often relies on precise oculomotor control ([30]; [31]).Consequently, the role of oculomotor maturation in shaping eye movement patterns during Chinese reading may be more significant than when reading alphabetic scripts.

To date, only one experimental study has directly investigated the drivers of children’s eye movement development in Chinese reading ([35]). The finding suggested that both linguistic abilities and oculomotor maturation contribute to the development of eye movement patterns in Chinese reading, supporting the combined hypothesis. Under the assumption that children’s oculomotor maturation was consistent within the same age group, the researchers used physiological age as a proxy for oculomotor maturation and text reading comprehension as an indicator of linguistic abilities. The results showed a significant interaction between age and reading comprehension in influencing children’s eye movement patterns. Specifically, reading comprehension exerted the greatest influence on eye movement development in 9-year-olds, followed by 10-year-olds, with this effect diminishing by age 11. The study concluded that the development of reading eye movement patterns is driven by the interaction between improvements in linguistic abilities and the maturation of oculomotor control.

However, the study by [35] ([35]) raises two issues that require further consideration. First, the reliance on chronological age as the primary indicator of oculomotor maturation is problematic, as age is inherently correlated with the development of other cognitive abilities, including linguistic abilities ([20], [19]; [46]; [47]; [65], [64]). Consequently, oculomotor maturation cannot be disentangled from the influence of chronological age alone. To address this limitation, it is essential to employ direct measures of oculomotor maturation to better understand its development in children. The saccade task (ST) offers an effective method for examining oculomotor maturation ([22]; [27]; [28]; [29]). In this paradigm, participants fixate on a central point and are instructed to perform either a prosaccade (a forward saccade toward the same direction as a peripheral target stimulus) or an antisaccade (an inhibitory saccade in the opposite direction of the target stimulus). The prosaccade task assesses the ability to disengage visual attention, while the antisaccade task evaluates inhibitory control over eye movements ([1]). [28] ([28]), as well as [29] ([29]) employed this paradigm to investigate the development of oculomotor maturation in children and found that adults exhibited faster saccade latencies than children in both tasks. Additionally, the accuracy of saccade landing positions increased with age, while the error rates in inhibitory saccades decreased ([7], [8]). Similarly, [22] ([22]) used this paradigm to measure oculomotor maturation when examining its influence, alongside linguistic abilities, on children’s reading eye movement patterns in alphabetic scripts. They concluded that oculomotor maturation did not significantly predict fixation durations in reading tasks.

The second issue with [35] ([35])’s study lies in their use of text reading comprehension as a measure of linguistic abilities. Reading comprehension is a complex cognitive process that involves not only basic reading skills, such as lexical decoding, but also higher-order linguistic skills, including inference-making, integration, reading monitoring, and awareness of text structures ([51]; [61]). Therefore, it is necessary to further investigate how specific subcomponents of reading comprehension influence the development of children’s eye movement patterns. During the elementary school years, children transition between the stages of “learning to read” and “reading to learn” ([10]). Their primary tasks involve acquiring decoding skills and linguistic comprehension, which are foundational components of linguistic abilities ([17]; [21]; [52]). Researchers commonly use Chinese character recognition to assess decoding skills and vocabulary knowledge to measure linguistic comprehension ([52]; [68]; [69]). Therefore, the present study focuses on these two components—Chinese character recognition and vocabulary knowledge—to clarify the linguistic factors driving the development of children’s eye movement patterns during reading.

The current study investigates the driving factors behind the development of eye movement patterns in Chinese children’s reading. Saccade tasks were used to measure oculomotor maturation, while Chinese character recognition and vocabulary knowledge assessed linguistic abilities. By examining the predictive roles of oculomotor maturation and linguistic abilities on children’s reading eye movement measures, this study aims to uncover the mechanisms driving the development of eye movement patterns in Chinese readers.

We hypothesize that if children’s eye movement measures are predicted solely by linguistic abilities, the findings would support the linguistic-proficiency hypothesis. Conversely, if oculomotor maturation alone predicts eye movement measures, this would provide evidence for the oculomotor-tuning hypothesis. More importantly, if the interaction between linguistic abilities and oculomotor maturation significantly predicts the development of eye movement measures, the results would support the combined hypothesis.

## 2. Method

### 2.1. Participants

This study was approved by the Human Research Ethics Committee of Tianjin Normal University. Informed consent was obtained from the participating school and parents prior to data collection. The sample consisted of 215 children from first to fifth grade, recruited from an elementary school in Tianjin, China. Participants included 40 children in first grade (17 girls, *M*_age_ = 6.83 years, *SD*_age_ = 0.39 years), 42 in second grade (18 girls, *M*_age_ = 7.93 years, *SD*_age_ = 0.31 years), 45 in third grade (20 girls, *M*_age_ = 8.79 years, *SD*_age_ = 0.42 years), 43 in fourth grade (20 girls, *M*_age_ = 9.76 years, *SD*_age_ = 0.44 years), and 45 in fifth grade (16 girls, *M*_age_ = 10.79 years, *SD*_age_ = 0.43 years).

All participants had normal or corrected-to-normal vision and passed the reading fluency test and Raven’s progressive matrices, confirming the absence of reading disabilities or cognitive impairments. Teachers verified that all children were typically developing with no diagnosed developmental deficits. Upon completing the study, each child received a stationery gift valued at RMB 20. Additionally, 40 university students (24 females, *M_age_* = 20.40 years, *SD_age_* = 1.72 years) were recruited as an adult control group.

### 2.2. Tasks and Instruments

Reading eye movement tasks, saccade tasks, and linguistic ability tests were used. All tasks were conducted individually in a quiet room in their school. Three trained experimenters administered the tests one-on-one with the participants in two days. Half of the participants in each grade first performed the reading eye movement task, followed by the saccade tasks on the first day, and then took the linguistic ability tests (the Chinese character recognition task followed by the vocabulary knowledge test) on the second day. The other half began with the linguistic ability tests on the first day and then completed the reading eye movement task and saccade tasks on the second day. Within each day of tasks, the order was fixed. The total testing time for each participant did not exceed 80 min.

#### 2.2.1. Reading Eye Movement Task

Reading materials: The materials were selected and refined through a multi-step process. Sentences were initially drawn from Chinese language textbooks published by Beijing Normal University Press and Jiangsu Education Press. Ten Chinese language teachers evaluated the sentences for difficulty and fluency using a 5-point Likert scale (5 = “very difficult”/“not fluent”, 1 = “very easy”/“very fluent”). Based on their evaluations and recommendations, the sentences were revised and further screened. Subsequently, five students from each grade from first to fifth grade individually assessed the difficulty of the sentences. Final selections were made by combining the teachers’ and students’ evaluations to ensure the materials matched participants’ reading proficiency (see Table 1 for details). Each grade was provided with 20 age-appropriate sentences, including 4 practice sentences and 16 experimental sentences. To control for differences in material difficulty across grades, children in second to fifth grade also read a set of sentences designed for first -graders as common stimuli.

Apparatus and procedure: Eye movements were recorded using an Eyelink 1000 eye tracker with a sampling rate of 1000 Hz. Stimuli were displayed on a 19-inch Dell LCD monitor with a refresh rate of 144 Hz and a resolution of 1980 × 1080 pixels. Participants were seated 65 cm from the monitor. Before the experiment, instructions were provided, followed by a three-point calibration procedure to achieve an accuracy of less than 0.4°. Upon successful calibration, participants began the formal experiment.

During the task, participants focused on a central black-and-white circular fixation point on the left side of the screen. Sentences were presented centrally, with each Chinese character rendered at 48 pixels, approximately 1.1° of visual angle. Participants read each sentence silently and pressed a key to proceed to the next sentence. One-third of the sentences were followed by a comprehension question, which participants answered via a key press before continuing. This task was completed within 20 min.

#### 2.2.2. Saccade Tasks (ST)

The same eye tracker used in the reading eye movement task was employed for the saccade tasks. A central fixation point, represented by a green square (0.5 × 0.5 cm), was displayed on a black background, with its duration randomly varying between 2000 and 3500 ms. Subsequently, a peripheral target, represented by a red square (0.5 × 0.5 cm), appeared 24 cm to the left or right of the central fixation point and remained visible for 1000 ms. Participants were instructed to shift their fixation as quickly as possible to the target’s location.

The experiment included two tasks, namely the prosaccade task (PST) and the antisaccade task (AST). For the PST, participants performed three sub-tasks: (1) in the overlap task, the red central fixation point remained visible for 1000 ms after the peripheral green target appeared; (2) in the step task, the red central fixation point disappeared as soon as the peripheral green target appeared; (3) in the gap task, the red central fixation point disappeared 200 ms before the appearance of the peripheral green target. In the AST, the peripheral target appeared in the same manner as in the gap task, but participants were required to shift their fixation to a position opposite to the target direction but equidistant from the central fixation point.

The experimental procedure is illustrated in Figure 1. Participants completed a practice session before the formal experiment to ensure they fully understood the task requirements. Each task consisted of a block of 16 trials, lasting approximately 90 s. The total duration of all these tasks was about 20 min, with the presentation order randomized across participants.

#### 2.2.3. Linguistic Ability Tests

The linguistic ability tests consisted of the Chinese character recognition task and the vocabulary knowledge test. Both tests were administered sequentially and completed within a total duration of 40 min.

In the Chinese character recognition task, participants were presented with 150 Chinese characters arranged in ascending order of difficulty. They were instructed to read the characters aloud, earning one point for each correct response. Testing was terminated after 15 consecutive errors ([32]). The internal consistency of this test, as measured by Cronbach’s α, was 0.91.

In the vocabulary knowledge test, participants were orally presented with a word and asked to explain its meaning. Their responses were recorded verbatim by the experimenter and later rated on a 0–2 scale by two independent raters based on semantic appropriateness. This test included 32 items arranged in ascending order of difficulty, and testing was terminated after 5 consecutive scores of 0 ([73]). The internal consistency of this test, as measured by Cronbach’s α, was 0.89.

### 2.3. Data Analysis

Data analysis was conducted in three stages. First, data from the reading eye movement task and saccade tasks were preprocessed to exclude fixation points and saccades that did not meet the experimental criteria. Second, grade differences in the reading eye movement task, saccade tasks, and linguistic ability tests were analyzed using R (version 4.4.2). This included both adjacent grade comparisons and pairwise comparisons across all grades, with statistically significant group differences reported. Finally, path analysis was used to explore how oculomotor maturation and linguistic abilities predict eye movement patterns to reveal the mechanisms underlying children’s reading development.

#### 2.3.1. Data Filtering

In the reading eye movement task, the average reading comprehension accuracy was 91% across all participants (first-graders: *M* = 86%, *SD* = 0.13; second-graders: *M* = 88%, *SD* = 0.12; third-graders: *M* = 92%, *SD* = 0.14; fourth-graders: *M* = 94%, *SD* = 0.06; fifth-graders: *M* = 92%, *SD* = 0.09; adults: *M* = 95%, *SD* = 0.05), indicating that participants read and comprehended the experimental sentences attentively. Data exclusions were applied using the following criteria: (1) fixations shorter than 80 ms or longer than 1200 ms; (2) sentences incompletely read due to experimental interruptions or sudden disappearance of fixation points; and (3) sentences with fewer than five fixations. Overall, 2.39% of the data were excluded.

In saccade tasks, areas of interest (AOIs) were defined as 1.4 × 1.4 cm squares centered on the central fixation point and peripheral targets, dividing the screen into two equally sized AOIs (25.1 × 14.5 cm each) on the left and right sides. Consistent with prior studies ([16]; [72]; [74]), only first saccade data were analyzed, and exclusions were based on (1) blinks occurring during saccades; (2) starting points deviating more than 1° of visual angle from the central fixation point; and (3) saccades with latencies shorter than 80 ms or longer than 700 ms. Overall, a total of 11.79% of the data were excluded.

#### 2.3.2. Analysis of Developmental Characteristics

In the reading eye movement task, two types of measures were analyzed, namely temporal measures represented by mean fixation duration, and spatial measures represented by forward saccade amplitude ([46]; [47]). To evaluate the grade developmental characteristics of these measures, linear mixed-effects models were employed using the lme4 package in R ([4]). The initial model included maximal random effects, treating grade, and sentence length as fixed effects and accounted for random effects of participants and items. If the maximal model failed to converge, a zero-correlation parameter model was used, removing random effect terms with minimal variance to achieve convergence ([3]). Grade differences in eye movement measures were then analyzed based on these model results.

In the saccade tasks, the gap, step, overlap, and antisaccade tasks were used to evaluate the percentage of erroneous saccades, percentage of gain, and latency effect ([1]). The percentage of erroneous saccades was calculated as the proportion of trials with incorrect saccade directions. The latency effect for directionally correct saccades was analyzed using the step task as a baseline. The gap effect was the latency delta between the gap and step tasks, reflecting the ability to disengage from a fixation and rapidly reorient visual attention. The overlap effect was the latency delta between the overlap and step tasks, reflecting the ability to inhibit fixation on a current stimulus, disengage from a target, and reorient attention. The inhibitory control effect was the latency delta between the antisaccade and gap tasks, reflecting inhibitory control over saccades. The percentage of gain was calculated as the ratio of actual saccade distance to the theoretical distance required to reach the peripheral target during correct saccades.

To examine sources of variation in oculomotor maturation, we conducted an exploratory principal component analysis (PCA), according to [15] ([15]) and [11] ([11]). Due to the high percentage of erroneous saccades in the antisaccade task, 74 participants lacked data for the percentage of gain and latency effect in this task. Consequently, we focused on the remaining 10 measures and analyzed them using the psych package in R, applying an oblique (Oblimin) rotation to allow for correlations between factors. We excluded measures with a KMO value below 0.6, factor loadings below 0.4, or cross-loadings above 0.4. Two factors were ultimately extracted, explaining 85.20% of the total variance: the first factor accounted for 59.80%, and the second for 25.50%. Variance analyses and post hoc tests were conducted on these two factors to assess grade developmental trends in percentage of gain for forward saccades and percentage of erroneous saccades for the antisaccade task.

For the linguistic ability tests, one-way ANOVAs were conducted with grade as the independent variable to analyze total scores on the Chinese character recognition task and vocabulary knowledge test. Tukey’s HSD post hoc tests were applied for significant ANOVA results, to evaluate the grade developmental trends in Chinese character recognition and vocabulary knowledge.

#### 2.3.3. Analysis of Driving Factors

To investigate the driving factors underlying the development of children’s reading eye movement patterns, standardized scores from the Chinese character recognition and vocabulary knowledge tests were used as indicators of linguistic abilities. Similarly, standardized scores for the percentage of erroneous saccades from the antisaccade task and the percentage of gain from the prosaccade tasks, derived through exploratory principal component analysis, represented oculomotor maturation.

We then conducted a path analysis using the R package *lavaan* to examine the predictive roles of linguistic abilities and oculomotor maturation, as well as their interaction effects, on eye movement measures. Mean fixation duration and forward saccade amplitude served as dependent variables, with a covariance term included to account for shared variance between these measures. Age was included as a control variable in the path analysis models. This analysis aimed to clarify how linguistic abilities and oculomotor maturation independently and interactively contribute to the development of children’s eye movement patterns.

To verify the robustness of the path analysis results, we conducted supplementary analyses using frequentist LMM (lmerTest package) and Bayesian mixed effects models (brms package). Mean fixation duration and forward saccade amplitudes were modeled as dependent variables, with linguistic abilities, oculomotor maturation, and their interaction as fixed effects. Random effects included participant-level intercepts and slopes for linguistic abilities, and item-level intercepts. Bayesian models used weakly informative priors (normal(0,5)) and were estimated using 5 Markov chains with 5000 iterations each. Convergence was assessed using Rhat, trace plots, and posterior predictive checks.

## 3. Results

### 3.1. Developmental of Reading Eye Movement Patterns

As children age, mean fixation duration decreases (Figure 2, left), and forward saccade amplitudes increase (Figure 2, right). Significant developmental changes in mean fixation duration were observed between first and third grade, while forward saccade amplitudes showed significant development between second and third grade. Beyond third grade, the development of both eye movement measures plateaued, but significant differences remained between fifth grade children and adults.

Mean fixation duration decreased significantly with grade. Pairwise comparisons between adjacent grades revealed that first-graders exhibited significantly longer fixation durations than second-graders (*b* = 48.33, *SE* = 8.23, *t* = 5.82, *p* < 0.001, 95% CI [32.20, 64.50]), and second-graders had significantly longer durations than third-graders (*b* = 33.53, *SE* = 7.87, *t* = 4.26, *p* < 0.001, 95% CI [18.10, 49.00]). No significant differences were found between third and fourth grade (*b* = 6.02, *SE* = 7.53, *t* = 0.80, *p* = 0.424, 95% CI [−8.73, 20.80]) or fourth and fifth grade (*b* = 4.73, *SE* = 7.79, *t* = 0.61, *p* = 0.54, 95% CI [−10.55, 20.00]). However, fifth-graders demonstrated significantly longer fixation durations than adults (*b* = 36.17, *SE* = 7.98, *t* = −4.68, *p* < 0.001, 95% CI [20.50, 51.81]). Pairwise comparisons across all age groups revealed significant differences between most groups (*|b|* ≥ 33.53, *t* > 4.26, *p* < 0.001), except among third, fourth, and fifth grade, where differences were not significant (*|b|* ≤ 12.45, *t* ≤ 1.60, *p* > 0.05).

Forward saccade amplitudes increased significantly with grade. Comparisons between adjacent grades revealed no significant difference between first and second grade (*b* = −0.01, *SE* = 0.08, *t* = −0.12, *p* = 0.90, 95% CI [−0.12, 0.14]). However, third-graders showed significantly larger forward saccade amplitudes than second-graders (*b* = −0.20, *SE* = 0.08, *t* = −2.65, *p* = 0.008, 95% CI [−0.35, −0.05]). No significant differences were found between third and fourth grade (*b* = −0.05, *SE* = 0.08, *t* = −0.69, *p* = 0.49, 95% CI [−0.20, 0.09]) or fourth and fifth grade (*b* = −0.07, *SE* = 0.08, *t* = −0.88, *p* = 0.38, 95% CI [−0.20, 0.10]). fifth-graders exhibited significantly smaller forward saccade amplitudes than adults (*b* = −0.44, *SE* = 0.08, *t* = −5.65, *p* < 0.001, 95% CI [−0.60, −0.29]). Pairwise comparisons across all groups showed that first and second grade had significantly smaller forward saccade amplitudes than third, fourth, fifth grade, and adults (*|b|* ≥ 0.20, *|t|* ≥ 2.65, *p* < 0.05). All children exhibited significantly smaller forward saccade amplitudes than adults (*|b|* ≥ 0.44, *|t|* ≥ 5.65, *p* < 0.001), while no significant differences were found among third, fourth, and fifth grade (*|b|* < 0.12, *|t|* < 1.58, *p* > 0.05).

### 3.2. Developmental of Linguistic Abilities

Children’s performance in Chinese character recognition (Figure 3, left) and vocabulary knowledge (Figure 3, right) showed significant yearly improvements before fourth grade, with no notable development observed between fourth and fifth grade. Nevertheless, both abilities in fifth grade remained significantly lower than those of adult readers. The specific statistical comparisons are as follows.

For character recognition, the main effect of grade was significant (*F* (5, 249) = 127.90, *p* < 0.001, *η*^2^ = 0.72, Figure 3, left). Post hoc tests showed significant increases between first and second grade (Δ*M* = 26.42, *p* < 0.001, 95% CI [13.69, 39.13]), second and third grade(Δ*M* = 29.56, *p* < 0.001, 95% CI [17.20, 41.91]), and third and fourth grade (Δ*M* = 14.27, *p* = 0.01, 95% CI [1.98, 26.54]). No significant difference was observed between fourth and fifth grade (Δ*M* = 4.04, *p* = 0.93, 95% CI [−12.48, 20.56]), while a significant increase was found between fifth grade and adults group (Δ*M* = 23.99, *p* < 0.001, 95% CI [11.48, 36.50]). Pairwise comparisons across all age groups revealed significant differences between all groups except fourth and fifth grade (Δ*M*s > 14.27, *p*s < 0.01).

For vocabulary knowledge, the main effect of grade was significant (*F* (5, 249) = 177.30, *p* < 0.001, *η*^2^ = 0.78, Figure 3, right). Post hoc tests showed significant increases between first and second grade (Δ*M* = 4.54, *p* = 0.005, 95% CI [0.94, 8.15]), second and third grade (Δ*M* = 6.64, *p* < 0.001, 95% CI [3.14, 10.14]), and third and fourth grade (Δ*M* = 3.76, *p* = 0.03, 95% CI [0.28, 7.24]). No significant difference was observed between fourth and fifth grade (Δ*M* = −1.10, *p* = 0.94, 95% CI [−4.58, 2.38]), while a significant increase was found between fifth grade and adults group (Δ*M* = 20.67, *p* < 0.001, 95% CI [17.21, 24.21]). Pairwise comparisons across all age groups revealed significant differences between most groups except third and fifth grade, as well as fourth and fifth grade (Δ*M*s > 3.76, *p*s < 0.05).

### 3.3. Developmental in Saccade Task Performance

The results from the prosaccade and antisaccade Tasks (Figure 4) indicate that children’s abilities to disengage attention from a target and inhibit saccades improved with grade level. By fifth grade, their disengagement ability was comparable to that of adults, while their inhibitory control over saccades remained underdeveloped. Detailed statistical results are presented below.

For the percentage of erroneous saccades, significant grade differences were only observed in the antisaccade task (*F* (5, 249) = 13.83, *p* < 0.001, *η*^2^ = 0.22). Specifically, readers in first grade had significantly higher error rates than readers in third and above (|Δ*M*|s > 0.15, *p*s ≤ 0.05). All children exhibited significantly higher error rates than adults (|Δ*M*|s > 0.22, *p*s ≤ 0.001). However, no significant differences were observed between adjacent grades or among other groups (|Δ*M*|s < 0.14, *p*s > 0.05).

For the latency effect, significant grade differences were found only in the overlap effect (*F*s = 4.11, *p*s = 0.001, *η*^2^ = 0.08). Specifically, all children from first to fourth grade had significantly smaller overlap effects than adults (|Δ*M*|s > 28.53, *p*s < 0.05), while no significant differences were observed between adjacent grades or among other groups (|Δ*M*|s < 24.45, *p*s > 0.05).

For the percentage of gain, significant grade effects were observed across all four saccade tasks (*F*s > 2.75, *p*s < 0.05). For the three prosaccade tasks, first to third grade showed significant or marginally significant differences compared to adults group (|Δ*M*|s > 0.05, *p*s < 0.06). However, no significant differences were found between adjacent grades or among other groups (|Δ*M*|s < 0.04, *p*s > 0.05). For the antisaccade task, first grade showed significant differences compared to third fifth grade and adults group (Δ*M*s > 0.14, *p*s < 0.02), while no significant differences were observed between adjacent grades or among other groups (|Δ*M*|s < 0.12, *p*s > 0.05).

To further analyze the structure of oculomotor maturation, an exploratory principal component analysis (PCA) was conducted with standardized scores. The results showed that oculomotor maturation consisted of two independent factors. The first factor was the percentage of gain in the prosaccade tasks, reflecting the ability to disengage from a target ([26]). The second factor was the percentage of erroneous saccades in the antisaccade task, reflecting the ability to inhibit saccades ([49]). The developmental trends of these two factors are shown in Figure 4.

For the percentage of gain in the prosaccade tasks, first to fourth graders showed significantly lower scores than adults group (|Δ*M*|s > 0.05, *p*s < 0.05), but no significant differences were found between adjacent grades or among other groups (|Δ*M*|s < 0.03, *p*s > 0.05). For the percentage of erroneous saccades in the antisaccade task, first graders exhibited significantly higher error rates than readers in third grade and above (|Δ*M*|s > 0.15, *p*s ≤ 0.05). All children had significantly higher error rates than adults (|Δ*M*|s > 0.22, *p*s ≤ 0.001). However, no significant differences were observed between adjacent grades or among other groups (|Δ*M*|s < 0.14, *p*s > 0.05).

### 3.4. Prediction of Reading Eye Movement Patterns

The path analysis results (Figure 5) demonstrated distinct contributions of linguistic abilities and oculomotor maturation to reading eye movement patterns. When mean fixation duration was the predicted variable, linguistic abilities had a significant negative predictive effect (*β* = −0.67, *p* < 0.001), indicating that stronger linguistic abilities were associated with shorter fixation durations. In contrast, oculomotor maturation showed no significant predictive effect (*β* = −0.08, *p* = 0.11), and the interaction term between linguistic abilities and oculomotor maturation was also not significant (*β* = 0.07, *p* = 0.21).

When forward saccade amplitude was the predicted variable, linguistic abilities had a significant positive predictive effect (*β* = 0.54, *p* < 0.001), indicating that readers with stronger linguistic abilities exhibited larger forward saccade amplitudes. Oculomotor maturation had a marginally significant predictive effect (*β* = 0.10, *p* = 0.08), suggesting that readers with more developed oculomotor maturation tended to have slightly larger forward saccade amplitudes. Most notably, the interaction between linguistic abilities and oculomotor maturation was significant (*β* = 0.14, *p* = 0.02).

To further understand how linguistic abilities and oculomotor maturation interact to influence forward saccade amplitude, participants were categorized into high and low groups for each factor based on standardized means, and simple effects analyses were conducted. The results revealed that linguistic abilities had a stronger effect on forward saccade amplitude for readers with higher oculomotor maturation compared to those with lower oculomotor maturation (Δ*M*_low_ = 0.31, *t*_low_ = 4.40, *p*_low_ < 0.001, Cohen’s *d*_low_ = 0.95; Δ*M*_high_ = 0.41, *t*_high_ = 6.02, *p*_high_ < 0.001, Cohen’s *d*_high_ = 1.20).

To verify the robustness of the path analysis findings, supplementary analyses were conducted using frequentist LMM and Bayesian mixed effects models, with all model convergence parameters achieving Rhat ≈ 1.00, indicating proper chain mixing and convergence. Effective sample sizes (Bulk_ESS and Tail_ESS) exceeded 1000 for all key parameters, ensuring stable posterior estimates. In the mean fixation duration (MFD) model, the analysis revealed a significant main effect of linguistic abilities (*b* = −0.16, SE = 0.01, *t* = −13.08, *p* < 0.001, 95% CI [−0.19, −0.14]), indicating that higher language proficiency was associated with shorter fixation duration. The main effect of oculomotor maturity remained marginally significant (*b* = −0.02, SE = 0.01, *t* = −1.69, *p* = 0.09, 95% CI [−0.05, 0.01]). However, the interaction effect was non-significant (*b* = 0.03, SE = 0.02, *t* = 1.28, *p* = 0.20, 95% CI [−0.01, 0.06]). In the forward saccade amplitude model, linguistic abilities showed a robust positive association (*b* = 0.19, *SE* = 0.02, *t* = 8.44, *p* < 0.001, 95% CI [0.26, 0.33]), with oculomotor maturity demonstrating a significant main effect (*b* = 0.04, *SE* = 0.02, *t* = 2.03, *p* = 0.04, 95% CI [0.00, 0.09]). The interaction term exhibited a marginal significance (b = 0.07, *SE* = 0.04, *t* = 1.87, *p* = 0.06, 95% CI [0.00, 0.13]).

All the results indicate that linguistic abilities consistently modulate both temporal and spatial aspects of eye movement patterns during reading, whereas oculomotor maturity primarily influences saccadic amplitude control, albeit with weaker robustness.

## 4. Discussion

This study investigated the developmental trajectories of eye movement patterns in Chinese primary school children, and examined the contributions of linguistic abilities and oculomotor maturation to these patterns. The findings revealed the following: (1) Fixation durations decreased and saccade amplitudes increased with grade, yet neither measure reached adult levels by fifth grade. (2) Linguistic abilities improved significantly across grades but remained below adult levels. (3) Oculomotor maturation showed steady improvement, with disengagement abilities reaching adult levels by fifth grade, while saccade inhibition remained immature. (4) Temporal eye movement measures were primarily predicted by linguistic abilities, whereas spatial measures were influenced by both linguistic abilities and oculomotor maturation. These results highlight the distinct roles of linguistic and oculomotor development in shaping children’s reading eye movement patterns.

### 4.1. Development of Reading Eye Movement Patterns

The eye movement patterns of child readers exhibited a trend of shorter fixation durations and longer saccade amplitudes with increasing grades, with the most pronounced changes occurring between first and third grade. No significant differences were observed between third and fifth grade, yet fifth grade children still differed significantly from adult readers. These findings align with prior research ([5]; [20]; [46]; [47]). For instance, in their review of reading eye movement development in alphabetic scripts, Blythe and Joseph observed consistent decreases in sentence reading times and fixation durations, alongside increases in saccade amplitudes with age. Similarly, [47] ([47]) found that German-speaking children exhibited steady improvements in asymptotic reading performance across six grades, with the most pronounced gains in all measures occurring between first and third grade.

The significant reduction in fixation durations observed in the current study reflects an acceleration in information extraction and a higher degree of reading automatization ([58]; [59]). Concurrently, the increase in saccade amplitudes indicates enhanced visual information processing efficiency, enabling children to gather more information within a single fixation ([24]; [56]; [57]). As children progress through grades, their eye movement patterns increasingly demonstrate a more mature and efficient approach to information processing. This progression may highlight a developmental shift in their reading strategies, evolving from an initial character-by-character (or from letter-by-letter in alphabetic scripts) to a more advanced global processing strategy ([65]). This transition reflects children’s growing capacity to extract and integrate information more automatically as their reading skills develop ([13]; [14]).

These findings suggest that, in the context of Chinese reading, the temporal and spatial dimensions of eye movement patterns undergo a critical developmental period during the lower grades, followed by a more gradual and continuous phase of development. By fifth grade, these measures remain below adult levels, indicating a protracted developmental trajectory extending beyond primary school.

### 4.2. Development of Linguistic Abilities

The development of character recognition and vocabulary knowledge showed consistent improvement with increasing grades. Character recognition increased from an average of 44 characters in first grade to 119 characters in fifth grade, corresponding to 31% and 83% of adult level performance, respectively. Vocabulary knowledge increased from 11 points in first grade to 25 points in fifth grade, achieving 25% and 55% of adult level. Compared to character recognition, the maturity of vocabulary knowledge in fifth graders was lower.

These findings align with trends reported in previous studies ([68]; [69]), which attribute these patterns to the emphasis of early Chinese reading instruction on decoding skills, such as character recognition. Character recognition reflects the ability to map orthography to phonology and is a central focus of early reading instruction. As children advance through school, instructional priorities shift towards vocabulary knowledge, which involves not only character recognition but also semantic processing of the recognized characters ([52]).

The relative contributions of these linguistic skills development vary with age: character recognition exerts a stronger influence during early primary school years, whereas vocabulary knowledge becomes more impactful in the upper grades ([52]; [69]). Thus, this study suggests that Chinese primary school children achieve near-adult levels of character recognition relatively early, while their vocabulary knowledge development remains substantially below adult levels by fifth grade.

### 4.3. Development of Oculomotor Maturation

The development of oculomotor maturation in children showed task-dependent trends, reflecting differing developmental trajectories for basic attention-shifting abilities and more complex executive functions. In the antisaccade task, the percentage of erroneous saccades significantly decreased with age, indicating gradual improvement in inhibitory control. However, even by fifth grade, error rates remained higher than those for adults, suggesting that inhibitory control continues to mature beyond primary school years. This aligns with prior research indicating that, although the neural circuits for inhibitory control are established by age 6, younger children often struggle to effectively utilize these capabilities, resulting in poorer antisaccade performance ([27]; [28]; [49]). The findings emphasize the protracted developmental trajectory of inhibitory control, which remains underdeveloped in older children. In contrast, prosaccade tasks revealed no significant age-related differences in erroneous saccades, suggesting that the basic ability to disengage attention from a fixation point and accurately redirect it to a new target develops early and reaches adult-like levels in young children. However, the percentage of gain—a measure of saccade accuracy—improved steadily with age, stabilizing at adult levels by approximately fourth grade (around age 9). These findings are consistent with earlier studies showing that saccade accuracy stabilizes after age 8 ([48]; [60]). Nonetheless, more recent research has documented subtle age-related improvements in saccade accuracy beyond this point ([1]; [7], [8]), likely due to advancements in eye-tracking precision and larger sample sizes that allow for detecting finer developmental changes ([60]).

Overall, these results highlight that oculomotor maturation develops unevenly across tasks. While children’s attention disengagement abilities reach adult-like levels by fourth grade, the inhibitory control required for saccade inhibition continues to mature but remains underdeveloped in fifth grade. This divergence underscores the prolonged developmental trajectory of executive functions involved in eye movement control, particularly in tasks demanding inhibition.

### 4.4. Driving Factors in the Development of Reading Eye Movement Patterns

This study employed rigorous operationalization of oculomotor maturation and linguistic abilities to investigate their roles in shaping children’s reading eye movement patterns. The results revealed a clear dichotomy: temporal eye movement measures, such as fixation durations, were driven solely by linguistic abilities, whereas spatial measures, such as forward saccade amplitude, were co-influenced by linguistic abilities and oculomotor maturation. These findings confirm the pivotal role of linguistic development in reading eye movement patterns and provide support for the hypothesis regarding the development of spatial measures.

#### 4.4.1. Contributions of Linguistic Abilities

Linguistic abilities primarily accounted for the development of temporal eye movement measures. Enhanced linguistic skills allowed children to process textual information more efficiently, leading to shorter fixation durations and longer saccade amplitudes. These findings align with prior research on both alphabetic and logographic scripts ([22]; [35]; [58]). Improved character recognition facilitated the rapid integration of a character’s orthographic, phonological, and semantic properties, reducing the time required for word-level processing. Similarly, expanded vocabulary knowledge enhanced semantic integration, enabling quicker comprehension of sentence meaning and reducing fixation durations. This efficient lexical processing helped children to achieve cognitive thresholds for word recognition more quickly, triggering subsequent saccades ([58]).

Additionally, advanced linguistic abilities allowed children to better utilize contextual cues to predict upcoming words ([36]; [39]), further increasing reading efficiency. This predictive processing enabled larger saccade amplitudes, as children could achieve more information fixation ([37]; [38]; [40]). Parafoveal processing, a hallmark of mature reading strategies ([54]; [65]), further supported these trends. As children’s decoding and comprehension skills advanced, they allocated more cognitive resources to parafoveal regions ([56], [57]), allowing for the previewing of orthographic and semantic information ([2]; [12]; [38]), which reduced the next word’s fixation duration and enabled a longer saccade amplitude. These findings underscore the importance of linguistic abilities in driving the development of efficient temporal reading behaviors.

#### 4.4.2. Contributions of Oculomotor Maturation

Our study found that oculomotor maturation primarily influenced spatial measures, such as forward saccade amplitude. This finding diverges from [35] ([35]), who reported joint effects of linguistic abilities and oculomotor maturation on both fixation durations and saccade amplitudes. The discrepancy may stem from [35] ([35])’s use of chronological age as an indicator of oculomotor maturation, which inherently confounds the developmental effects of linguistic abilities ([20]; [55]). In contrast, the present study employed saccade tasks to directly assess oculomotor maturation, allowing us to isolate its contribution more precisely. Our results reveal that oculomotor maturation significantly predicts spatial measures but not temporal measures.

The result suggests that oculomotor maturation primarily influences the precision and stability of eye movements, rather than the cognitive processes underlying reading speed or comprehension. One plausible explanation is that oculomotor maturation is directly responsible for the accuracy and control of saccadic movements, ensuring precise fixation shifts to new visual targets ([29]; [53]). As this ability matures, children become more efficient at executing saccades, reducing errors caused by inaccurate eye jumps, and ultimately facilitating precise movement from one word or phrase to the next.

#### 4.4.3. Cross-Linguistic Comparisons: Driving Factors in Chinese vs. Alphabetic Scripts

Regardless of whether the script is Chinese or an alphabetic language, research consistently shows that temporal measures of eye movements are more closely linked to linguistic ability. These findings confirm the assumptions of major eye movement control theories ([34]; [58]), which propose that fixation duration primarily reflects the efficiency of cognitive processing, such as word recognition and semantic access, rather than the precision of saccadic control.

However, our findings reveal a key distinction in spatial measures. In Chinese reading, oculomotor maturation plays a significant predictive role, whereas in alphabetic scripts, its influence is highly limited ([58]; [22]). It is important to note that [58] ([58]) based their conclusions on computational modeling rather than empirical data, while [22] ([22]) focused solely on fixation duration and did not examine saccade amplitude. Despite these methodological differences, we argue that the structural and visual properties of Chinese and alphabetic scripts may contribute to the differential effects of oculomotor maturation.

A key factor underlying this divergence is the higher visual complexity of Chinese characters, which contain intricate strokes, compact spatial layouts, and holistic structures ([71], [70]). These features impose greater demands on visual processing and necessitate precise oculomotor maturation. A mature oculomotor system allows children to efficiently and accurately guide their saccades across complex Chinese text, facilitating fluent reading. In contrast, alphabetic scripts have linear text layouts with explicit word spacing, providing natural visual targets for saccades. As a result, in alphabetic scripts, saccade amplitude is primarily influenced by word length and linguistic factors rather than oculomotor maturation ([56]; [58]).

Neuroimaging studies further support these script-related differences, revealing not only that Chinese reading engages the same left-lateralized language areas as alphabetic script reading—such as the left ventral inferior frontal gyrus and left temporoparietal junction—but also that it significantly activates the right occipitotemporal regions and parietal lobe. These right-hemisphere areas are crucial for the complex visual analysis and spatial processing required by Chinese characters ([18]; [66], [67]). Notably, the right parietal lobe plays a key role in integrating the spatial positioning of strokes and guiding target localization during dynamic saccadic eye movements ([18]). These findings highlight the unique demands of Chinese orthography on both linguistic and oculomotor systems, necessitating their jointcombined contribution to spatial measures of reading behavior.

### 4.5. Limitations and Future Research

It is important to note that, while the path analysis and mixed effects models employed in this study effectively elucidated the predictive roles of linguistic abilities and oculomotor maturation in children’s eye movement measures, they do not establish causal relationships among these variables.

Both analytic approaches consistently highlight the strong predictive effect of linguistic abilities, suggesting that linguistic proficiency plays a dominant role in shaping both temporal and spatial aspects of eye movement patterns during reading. In contrast, the effect of oculomotor maturation was notable, albeit weaker. Despite this discrepancy, we remain cautious in drawing definitive conclusions about the joint influence of linguistic abilities and oculomotor maturation on reading eye movement development.

To further clarify the study’s conclusions, future research should employ longitudinal and experimental designs to explore driving factors of eye movement measures. Such approaches would provide deeper insights into the mechanisms driving the development of reading behaviors in children.

## 5. Conclusions

In summary, this study highlights the distinct roles of linguistic abilities and oculomotor maturation in the development of eye movement patterns in Chinese reading. It provides evidence for the linguistic-proficiency hypothesis, demonstrating that linguistic abilities primarily drive the development of temporal eye movement measures. For spatial measures, the findings validate the combined hypothesis. These conclusions partially diverge from predictions made by the E-Z Reader model, highlighting the unique dual contributions of linguistic processing efficiency and oculomotor maturation in Chinese reading. 

## Figures and Tables

**Figure 1 behavsci-15-00426-f001:**
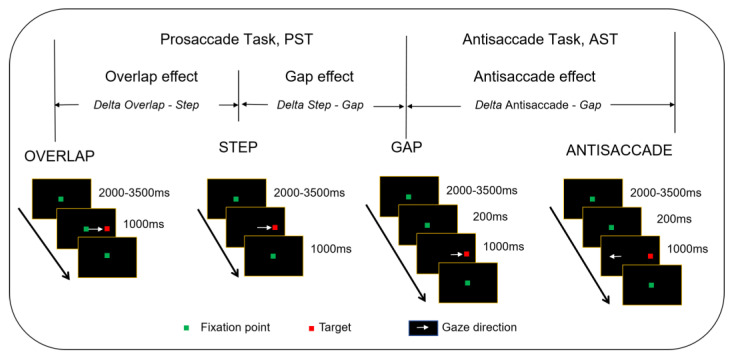
Experimental procedure of saccade tasks ([1]).

**Figure 2 behavsci-15-00426-f002:**
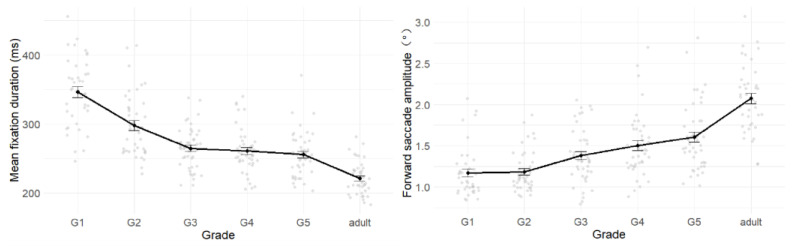
Development of reading eye movement patterns. *Note:* The **left** panel illustrates the mean fixation duration, and the **right** illustrates forward saccade amplitude.

**Figure 3 behavsci-15-00426-f003:**
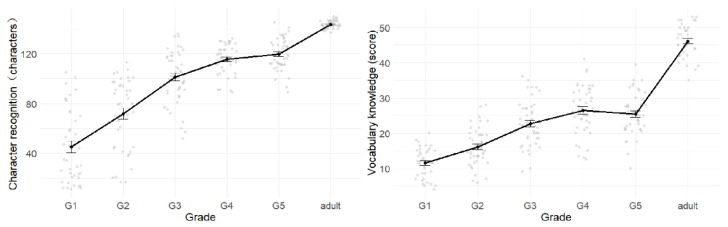
Development of linguistic ability. *Note:* The **left** panel illustrates Chinese character recognition, and the **right** panel illustrates vocabulary knowledge.

**Figure 4 behavsci-15-00426-f004:**
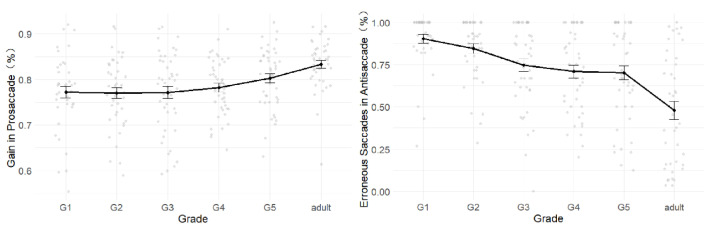
Development of oculomotor maturation measures. *Note:* The **left** panel illustrates gain of prosaccade, and the **right** illustrates the erroneous saccades of antisaccade.

**Figure 5 behavsci-15-00426-f005:**
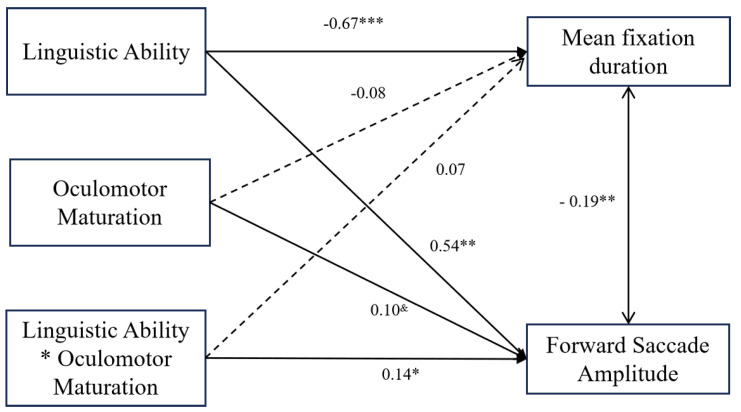
Prediction of reading eye movement patterns by linguistic abilities and oculomotor maturation. *Note:* * *p* < 0.05, ** *p* < 0.01, *** *p* < 0.001, ^&^
*p* < 0.10.

**Table 1 behavsci-15-00426-t001:** Difficulty, fluency, and examples of materials in the eye movement experiment.

Grade	Length (SD)	Difficulty (SD)	Fluency (SD)	Sentence Example
Grade 1	10.02 (0.87)	2.22 (0.38)	4.32 (0.48)	小鱼在水里玩游戏。 (Little fish are playing games in the water.)
Grade 2	11.46 (0.90)	2.47 (0.58)	4.25 (0.53)	家是我心中最美的地方。 (Home is the most beautiful place in my heart.)
Grade 3	13.75 (1.04)	2.31 (0.49)	4.28 (0.51)	中国是世界上最早制茶的国家。 (China is the first country to make tea in the world)
Grade 4	16.35 (0.86)	2.34 (0.43)	4.11 (0.48)	我们都知道水果和蔬菜中含有维生素。 (We all know that fruits and vegetables contain vitamins.)
Grade 5	19.49 (0.76)	2.43 (0.44)	4.15 (0.55)	竹子的品格体现了我们中华民族自强的精神。 (The character of bamboo embodies the spirit of self-improvement of our Chinese nation.)
Adults	23.00 (1.18)	2.21 (0.31)	4.04 (0.45)	他先后撰写了六篇法医论文在全国有关刊物上发表。 (He has authored six forensic papers published in relevant national journals.)

## Data Availability

The data presented in this study are available on request from the authors. The data are not publicly available due to privacy concerns.

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
