# Peer review of "Driving Factors in the Development of Eye Movement Patterns in Chinese Reading: The Roles of Linguistic Ability and Oculomotor Maturation"

_behavsci, 2025, doi:10.3390/bs15040426_

Round 1

Reviewer 1 Report

Comments and Suggestions for Authors

The current study conducted eye-tracking reading experiments, saccade tasks and linguistic abilities tasks to explore the mechanisms of the driving factors of the development of eye movement patterns in Chinese children. The experimental design and data analysis methods were solid and the results are convincing.

In general, I think this manuscript is well written and is qualified to be accepted. I believe it would provide great interests to the potential readers of Behavioral Sciences.

Author Response

Comments: I only have one minor concern as following: 

More discussions should be made regarding to the similar/different observations of the effects of linguistic abilities and oculomotor maturation between Chinese and alphabetic languages. This may shed some lights on whether script features would cause mutual/different reading behaviours and cognitive process, which may further strengthen the literature significance of the current study.

Response: We sincerely appreciate the reviewers' valuable suggestions. In our original manuscript, we discussed the differing effects of linguistic abilities and oculomotor maturation between Chinese and alphabetic languages within the section "Contributions of Oculomotor Maturation." However, this discussion was not explicitly emphasized. The reviewers' comments have made us recognize the importance of further highlighting this aspect.

To address this, we have introduced a dedicated subsection titled "Cross-linguistic Comparisons", where we provide a more in-depth discussion on these cross-language differences. The revised content can be found in the manuscript (pp. 642–679).

Cross-linguistic Comparisons: The Driving Factors in Chinese vs. Alphabetic Scripts

Regardless of whether the script is Chinese or an alphabetic language, research consistently shows that temporal measures of eye movements are more closely linked to linguistic ability. These findings confirm the assumptions of major eye movement control theories (Engbert et al., 2005; Li & Pollatse, 2020; Reichle et al., 2006), which propose that fixation duration primarily reflects the efficiency of cognitive processing, such as word recognition and semantic access, rather than the precision of saccadic control.

However, our findings reveal a key difference in spatial measures of eye movement. In Chinese reading, oculomotor maturation plays a predictive role, whereas in alphabetic scripts, its influence is highly limited (Reichle et al., 2013; Huestegge et al., 2009). It is important to note that Reichle et al. (2013) based their conclusions on computational modeling rather than empirical data, while Huestegge et al. (2009) focused solely on fixation duration and did not examine saccade amplitude. Despite these methodological differences, we argue that the structural and visual properties of Chinese and alphabetic scripts may contribute to the differential effects of oculomotor maturation.

A key factor underlying this divergence is the higher visual complexity of Chinese characters, which contain intricate strokes, compact spatial layouts, and holistic structures(Ming Yan et al., 2020; Yan et al., 2015). These features impose greater demands on visual processing and necessitate precise oculomotor maturation. A mature oculomotor system allows children to efficiently and accurately guide their saccades across complex Chinese text, facilitating fluent reading. In contrast, alphabetic scripts have linear text layouts with explicit word spacing, providing natural visual targets for saccades. As a result, in alphabetic scripts, saccade amplitude is primarily influenced by word length and linguistic factors rather than oculomotor maturation (Rayner, 1998; Reichle et al., 2003).

Neuroimaging studies further support these script-related differences, reveal that Chinese reading engages not only the same left-lateralized language areas as alphabetic script reading—such as the left ventral inferior frontal gyrus and left temporoparietal junction—but also significantly activates the right occipitotemporal regions and parietal lobe. These right-hemisphere areas are crucial for the complex visual analysis and spatial processing required by Chinese characters(Guo et al., 2022; Tan et al., 2005; Tan et al., 2000). Notably, the right parietal lobe plays a key role in integrating the spatial positioning of strokes and guiding target localization during dynamic saccadic eye movements(Guo et al., 2022). These findings highlight the unique demands of Chinese orthography on both linguistic and oculomotor systems, necessitating their combined contribution to spatial measures of reading behavior.

Reviewer 2 Report

Comments and Suggestions for Authors

The study aimed to tease apart the role of linguistic and oculomotor developmental factors on eye movements during reading through a series of experiments with Mandarin speaking children from grades 1 through 5 and found that linguistic development modulated eye movement fixation durations (how long children look at the words) while spatial measures ( the frequency and distance of eye movements) were modulated by both linguistic and oculomotor factors. The study fills in an important gap in the literature as indeed very few studies examine the developmental aspect of eye movements in reading by focusing on both oculomotor and linguistic variables. I recommend this work for the publication with Behavioral sciences journal pending a few minor revisions that I include below.

Results 3.4 Prediction of reading Eye Movement patterns sections – I was confused as to what kind of regression was run exactly. The authors call it “path analysis” but do not say what package in R was used – was it a typical regression or was a form of mixed-effects glm used with the independent variables as fixed factors and random factors as well? The reason I ask is a mixed effects model with random intercepts and slopes is preferable as it takes into account variability of slower or faster readers and harder versus easier items that are not accounted by the factors that the authors already included i.e. if Jonny participated in the morning and he is not a morning person or some of the sentences have words or  syntactic structures that seems harder or easier to your individual participants then what your 10 teacher-raters judged the sentences to be. (Barr, D. J., Levy, R., Scheepers, C., & Tily, H. J. (2013). Random effects structure for confirmatory hypothesis testing: Keep it maximal. Journal of memory and language, 68(3), 255-278.)

Comments on the Quality of English Language

It would be helpful to have a native speaker proofread the ms as there are some run ons and fragments. 

Author Response

Comments:I recommend this work for the publication with Behavioral sciences journal pending a few minor revisions that I include below.

Results 3.4 Prediction of reading Eye Movement patterns sections – I was confused as to what kind of regression was run exactly.  The authors call it “path analysis” but do not say what package in R was used – was it a typical regression or was a form of mixed-effects glm used with the independent variables as fixed factors and random factors as well?  The reason I ask is a mixed effects model with random intercepts and slopes is preferable as it takes into account variability of slower or faster readers and harder versus easier items that are not accounted by the factors that the authors already included i.e. if Jonny participated in the morning and he is not a morning person or some of the sentences have words or syntactic structures that seems harder or easier to your individual participants then what your 10 teacher-raters judged the  sentences to be.  (Barr, D. J., Levy, R., Scheepers, C., & Tily, H. J. (2013).  Random effects structure for confirmatory hypothesis testing: Keep it maximal.  Journal of memory and language, 68(3), 255-278.)

Response: We sincerely appreciate the reviewer’s insightful comments regarding the regression approach used in Section 3.4.

In our study, we employed path analysis using the R package lavaan. This approach was chosen to model the direct and indirect relationships between linguistic abilities, oculomotor maturation, and reading eye movement patterns. Given that both linguistic abilities and oculomotor maturation are continuous variables, path analysis provides a more comprehensive framework for simultaneously estimating multiple dependent variables, modeling their interrelationships. These content are elaborated in Section 2.3.3. of the manuscript (p347-352 ).

   We acknowledge the reviewer’s concern regarding individual differences in reading speed and item difficulty. While our current path analysis model does not include random intercepts and slopes.To further address the reviewer’s concerns, we also conducted a linear mixed-effects model (LMM) analysis after dichotomizing the independent variables (based on a 30% percentile split). Given that the log-transformed data exhibited a normal distribution and homoscedasticity, we opted for LMM rather than GLMM. The LMM approach included random intercepts to capture between-subject variability and random slopes where appropriate. Our findings remained consistent across both modeling approaches, supporting the robustness of our results. The detailed LMM results are provided in the supplementary materials at the end of the document.

We appreciate the reviewer’s constructive feedback and welcome any further suggestions for improving our analysis.

supplementary materials

TABLE :LMM of MFD and SAC

MFD

SAC

b

SE

t value

b

SE

t value

Intercept

393.69

35.35

11.137***

1.25

0.31

4.026***

Readability

-57.73

21.02

2.75**

-0.01

0.18

-0.06

Ocular

-2.43

20.57

-0.12

-0.28

0.18

-1.55

Readability:Ocular

-3.79

12.13

-0.31

0.22

0.11

2.13*
